# The Contribution of QF-PCR and Pathology Studies in the Diagnosis of Diandric Triploidy/Partial Mole

**DOI:** 10.3390/diagnostics11101811

**Published:** 2021-09-30

**Authors:** Leticia Benítez, Montse Pauta, Cèlia Badenas, Irene Madrigal, Alfons Nadal, Edda Marimon, Antoni Borrell

**Affiliations:** 1BCNatal, Department of Maternal-Fetal Medicine, Institute Gynecology, Obstetrics and Neonatology, Hospital Clínic de Barcelona and Hospital Sant Joan de Déu, 08028 Barcelona, Spain; lbenitez@clinic.cat (L.B.); memarimon@clinic.cat (E.M.); 2BCNatal, Institut d’Investigacions Biomèdiques August Pi i Sunyer (IDIBAPS), 08036 Barcelona, Spain; mpauta@clinic.cat; 3Biochemistry and Molecular Genetics Department, Hospital Clínic de Barcelona, 08036 Barcelona, Spain; cbadenas@clinic.cat (C.B.); imadbajo@clinic.cat (I.M.); 4Institut d’Investigacions Biomèdiques August Pi i Sunyer (IDIBAPS), 08036 Barcelona, Spain; 5Centre for Biomedical Research on Rare Diseases (CIBERER), ISCIII, 28029 Madrid, Spain; 6Department of Pathology, Hospital Clínic de Barcelona, 08036 Barcelona, Spain; anadal@clinic.cat; 7Department of Basic Clinical Practice, Universitat de Barcelona, 08036 Barcelona, Spain; 8Molecular Pathology of Inflammatory Conditions and Solid Tumors, Institut d’Investigacions Biomèdiques August Pi i Sunyer (IDIBAPS), 08036 Barcelona, Spain

**Keywords:** triploidies, molecular analysis, pathological analysis, QF-PCR, paternal origin

## Abstract

Objective: the aim of our study was to assess the contribution of quantitative fluorescent polymerase chain reaction (QF-PCR) and pathology studies in the diagnosis of diandric triploidies/partial hydatidiform moles. Methods: this study included all fet al triploidies diagnosed by QF-PCR in chorionic villi or amniotic fluid in the 2 centers of BCNatal in which a maternal saliva sample was used to establish its parental origin. Pathology studies were performed in products of conception and concordance between a partial hydatidiform mole diagnosis and the finding of a diandric triploidy was assessed. Results: among 46 fetal triploidies, found in 13 ongoing pregnancies and in 33 miscarriages, there were 26 (56%) diandric triploidies. Concordant molecular (diandric triploidy) and pathology results (partial mole) were achieved in 14 cases (54%), while in 6 cases (23%) pathology studies were normal, and in the remaining 6 cases (23%) pathology studies could not be performed because miscarriage was managed medically. Conclusions: diandric triploidy is associated with partial hydatidiform mole and its diagnosis is crucial to prevent the development of persistent trophoblastic disease. QF-PCR analysis in chorionic villi or amniotic fluid provides a more accurate diagnosis of the parental origin of triploidy than the classical pathology studies.

## 1. Introduction

Triploidy is the second most frequent chromosomal anomaly causing miscarriage, accounting for 10–13% of cases [1,2]. The genome in a triploidy consists of two maternal and one paternal (digynic), or two paternal and one maternal (diandric) set of chromosomes. In accordance with the 2/3 prevalence reported in the 80 s [3,4] it is well established that most triploidies are diandric in origin and caused by dispermy. Typically, the fetus with a diandric triploidy has relatively normal growth, with either a normal head size or microcephaly and the placenta usually has the appearance of a partial mole, although the majority of paternally derived cases result in earlier pregnancy loss. In digynic triploidy, fetuses are usually early growth-restricted, relatively macrocephalic and have a small placenta without molar changes. The mechanism of origin is fertilization of a diploid oocyte mostly due to a meiosis II error [5,6], and some are associated with a late demise involving a well-formed fetus.

Partial hydatidiform moles display moderate trophoblastic hyperplasia in a mixture of normal and edematous villi, and embryonic/fetal tissue may be present. In contrast, complete hydatidiform moles are characterized by the absence of embryo, trophoblastic hyperplasia and hydatidiform degeneration of all villi. Complete moles are mostly diploid and are caused by paternal uniparental disomy, while partial moles are associated with diandric triploidy [3].

Identification of molar pregnancies is crucial because they carry a 3.7% maternal risk for persistent gestational trophoblastic disease after a partial molar pregnancy. The macroscopic inspection of evacuated tissue and the histologic examination by Pathology studies are the classical methods applied to identify partial moles and have demonstrated a high specificity [7]. However, sensitivity is low and does not improve much with the addition of the level of β-hCG [7]. Ultrasound alone, at the time of the diagnosis of a non-viable pregnancy is also a poor predictor.

G-banding karyotyping has been classically applied to diagnose chromosomal anomalies in recurrent pregnancy losses, and while it can detect triploidy, the parental origin of chromosomes cannot be identified. Chromosomal Microarray Analysis by means of Comparative Genomic Hybridization cannot detect all triploidies. On the contrary, quantitative fluorescent polymerase chain reaction (QF-PCR), which is a rapid technique based on the amplification of polymorphic short tandem repeat loci, permits the identification of triploidy. Even though QF-PCR is not a new tool in the diagnosis of triploid pregnancies, since the use of this technique as the first-line method of genetic analysis after chorionic villi sampling or amniocentesis is well established in pregnancies at high risk for chromosomal anomalies, the investigation of the origin of the extra chromosome endowment is not common practice.

The use of QF-PCR in the diagnosis of molar pregnancies has been previously reported by large labs [8], but we are not aware of clinical studies assessing its feasibility in clinical practice. In the present study, we investigated the contribution of both, QF-PCR and pathology studies, in the diagnosis of diandric triploidies/partial moles.

## 2. Materials and Methods

### 2.1. Samples

This is a retrospective study including two cohorts of pregnancies: 46 triploid pregnancies, either ongoing or nonviable, diagnosed by QF-PCR and karyotype, and 36 additional pregnancies with a pathology diagnosis of partial mole with no QF-PCR studies, collected at the two BCNatal centers, Hospital Clinic Barcelona (HCB) and Hospital Sant Joan de Déu (HSJD), during a 5-year study period (2015–2019). Ongoing pregnancies of the first cohort of pregnancies were sampled, either by chorionic villi sampling or amniocentesis depending on gestational age, because of ultrasound anomalies or a high risk of aneuploidy. In BCNatal centers, women with a nonviable pregnancy are routinely offered chorionic villi sampling for QF-PCR and karyotyping before uterine content evacuation. Maternal saliva was retrieved for parental origin analysis, at the time of the sampling or after the diagnosis of triploidy. The study was approved by IRB of Hospital Clinic Barcelona (HCB/2014/6020), and written informed consent was obtained from all women.

In ongoing pregnancies scans were performed transabdominally using an Acuson Antares system (Siemens Medical Solutions, Malver, PA, USA) or a Voluson E6 (GE Medical System, Zipf, Austria). In nonviable pregnancies, ultrasound examination was performed with the use of a transvaginal probe and included the measurement of gestational sac diameters or the crown-rump length if an embryo was present for accurate dating of pregnancy.

Chorionic villi samples were collected in warmed RPMI 1640 medium (BioWhittaker, Cambrex, Landen, Belgium) and delivered to the laboratory within a few hours, evaluated and processed as described elsewhere. Amniotic fluid was evaluated and processed with our routine procedure. Genomic DNA was extracted from CV samples using the standard extraction protocol from QIAamp DNA Mini Kit (Qiagen, Hilden, Germany). For amniotic fluid and maternal buccal washes between 0.5 and 2 mL of sample were used, and DNA extraction was performed using the Instagene Matrix (Bio-Rad Laboratories Inc., Hercules, CA, USA), following the manufacturer’s instructions.

### 2.2. QF-PCR Analysis

Quantitative Fluorescent Polymerase Chain Reaction (QF-PCR) analysis encompassing chromosomes 13, 18, 21, X and Y was used to detect triploidies, complete androgenetic uniparental disomies and maternal cell contamination. QF-PCR was performed using the Devyser Compact kit (Devyser, Hägersten, Sweden), following manufacturer’s instructions. The kit amplifies 26 highly informative markers from chromosomes 13, 18, 21, X and Y. PCR products were analyzed with an ABI3130XL (Applied Biosystems, Foster City, CA, USA) and GeneMapper v3.5 software was used to analyze the results. Peak area ratios between 0.8 and 1.4 were considered normal, whereas ratios above and below these were interpreted as trisomy, and the presence of three alleles of equal peak area was also considered trisomy. The presence of a single peak was considered uninformative and a minimum of two concordant informative markers was required to give a result. Significant maternal contamination was ruled out with the use of microsatellite markers for chromosomes 13, 18 and 21 included in the QF-PCR kit applied to maternal saliva. The diagnosis of a triploidy by QF-PCR was followed by a confirmation karyotype. Evacuation of uterus content was performed surgically between 9–12 weeks and when molar changes were suspected, otherwise medical management was recommended. Pregnancy follow-up was obtained by reviewing the medical records.

### 2.3. Histopathology Analysis

Partial mole diagnoses were reviewed by both Pathology Departments of HCB and HSJD. Samples selected for histological examination were routinely formaldehyde-fixed and paraffin-embedded. 4 µm-thick sections were routinely stained with H&E. Villous and maternal tissues were dissected from formaldehyde-fixed, paraffin-embedded samples. Histological dysmorphic villi (irregular villous contour, trophoblastic stromal inclusions and villi dimorphism) were submitted for further genotyping. In these cases, DNA was extracted using the QIAamp DNA FFPE Tissue kit (QIAGEN, Hilden, Germany) following the manufacturer’s protocol, and amplified with the Mentype Chimera PCR amplification kit (Biotype Diagnostic GmbH, Dresden, Germany). PCR products were run in either a 3130 or 3130XL Avant Genetic Analyzer (ABI, Foster City, CA, USA), and analyzed with a Chimeris^TM^ Monitor 2.0 (Biotype Diagnostic GmbH, Dresden, Germany). Cases were classified as triploid or non-triploid according to previously published criteria. The origin of triploidy was determined based on a combined evaluation of allele ratios and the source of those alleles with sufficient polymorphism [9]

### 2.4. Statistical Analysis

Stata statistical software v.15 was used for statistical analysis of maternal age, gestational age, viability of pregnancy, type of sampling, type of management, parental origin and β-hCG level. Continuous variables (maternal age, gestational age and β-hCG level) were checked with Shapiro-Wilk test and described by mean and SD after verifying they follow a normal distribution. Categorical variables (viability of pregnancy, type of sampling, type of management, parental origin) were described as number of observations and relative frequency. A *p*-value of <0.05 was considered significant.

## 3. Results

The first cohort included 46 triploidies diagnosed by means of QF-PCR and karyotype, 33 (72%) of which were found in nonviable pregnancies, and 13 (28%) in ongoing pregnancies. The reason for sampling in ongoing pregnancies was abnormal ultrasound findings (*n* = 9) and a high risk of aneuploidy (*n* = 4). In nonviable triploid pregnancies, karyotyping was offered to women as part of our routine miscarriage protocol with the use of chorionic villi sampling or amniocentesis before uterine content evacuation. Mean maternal age was 33.9 years (SD 4.2), with no differences between ongoing and nonviable pregnancies. The vast majority of samples studied were chorionic villi (*n* = 42; 91%), and only a small proportion were amniotic fluid (*n* = 3; 6.5%) or fetal blood (*n* = 1; 2%). After a triploidy diagnosis in ongoing pregnancies, all couples requested a termination of pregnancy. Miscarriage was managed surgically in 19 (58%) pregnancies, and medically in the remaining 14 (42%), the latter precluding the performance of pathology studies because no products of conception were obtained.

Among the 46 triploidies studied from the first cohort, 20 (43%) were digynic (group A) and 26 (56%) were diandric (group B + C) (Figure 1). No differences were observed in maternal age between diandric (33.1 years) and digynic triploidies (35.2; *p* = 0.17), nor in β-hCG level, as detailed in Table 1. Gestational age differed between groups, with a mean gestational age of 8.2 weeks in the diandric, and 10.2 weeks in the digynic triploidies (*p*= 0.04), that can be related to the fact that only 8% of diandric triploidies were found in ongoing pregnancies, while this proportion was 55% in digynic triploidies (*p* = 0.001). Five additional chromosomal anomalies were found as well as the triploidy, 3 autosomal trisomies (70,XXY, +21; 70,XXY, +16; 70,XXY, +13), a mosaicism of trisomy 21 (69,XXX/70, XXX+21) and a double Y (70, XXYY). A single triploid case was found to be diallelic and therefore of mitotic origin, instead the remaining cases presented a triallelic pattern suggesting a different origin.

Among the 26 diandric triploidies, 12 lacked a confirmatory pathology report of a partial mole (group B), in six (23%) because the medical management of miscarriage precluded the obtention of products of conception, and a further six (23%) had an initial normal pathology report. The latter were reviewed by the senior perinatal pathologist, and three cases were reclassified to partial moles since they met the histopathology diagnostic criteria, and the remaining three were considered inconclusive, either because gestational age was too early, or due to a prolonged retention time of the products of conception. Finally, 14 diandric triploidies had a concordant pathology report of partial mole (group C).

Among the 20 digynic triploidies (group A), a partial mole was ruled out by pathology studies in six (36%), and in the remaining 14 (64%) cases no pathology studies were performed due to medical management of pregnancy loss. 

After reviewing the partial mole diagnoses at the Pathology Departments of HCB and HSJD, 36 additional cases were found not to have been previously studied by chorionic villi sampling and QF-PCR (Second cohort = group D). In 10 of these cases, the pathology diagnosis was confirmed by genotyping. Adding the 20 previous diandric triploidies with QF-PCR and pathology studies (group C+ non-confirming pathology cases of group B), there were 56 partial moles with a pathology report. Overall, the sensitivity of pathology was 90% (44/49), decreasing to 72% (13/18) when only QF-PCR ascertained cases were considered. There was a single failed QF-PCR study after chorionic villi sampling leading to a 4% (1/26) failure rate. We detailed the characteristics of each group in Table 2.

## 4. Discussion

Diandric triploidy is associated with partial hydatidiform mole and its diagnosis is crucial to identify persistent trophoblastic disease which accounts for 3.7% of the cases, according to a large review that included 2651 partial moles [10]. As far as diandric triploidy/partial mole is concerned, two taxonomies—one genetic (diandric triploidy) and one anatomopathological (partial mole) are in use. This study was conducted to clarify the concordance between the two taxonomic systems. QF-PCR analysis in both chorionic villi/amniotic fluid and in parental samples provided an accurate diagnosis of the parental origin of triploidy and appears to be more accurate than the classical pathology studies. Among the 46 fetal triploidies with QF-PCR included in the study, 26 were diandric and expected to present with partial mole changes at pathology studies. Concordant molecular and pathology results were observed in 14 cases (54%), while in six cases (23%) the results were discordant with normal pathology studies, and in the remaining six cases (23%) the medical management of miscarriage precluded the analysis of the products of conception.

Even though the association between diandric triploidy and partial mole is well established, few cases of diandric triploidies have been shown not to develop a partial mole phenotype [5], probably due to the low gestational age. Furthermore, few cases of partial mole were shown to present with a different chromosomal anomaly than triploidy, such as tetraploidy, autosomal trisomies or mosaicisms with a dominance of the paternal genome [11]. A large study conducted in a Canadian lab analyzed over 400 cases of suspected placental mole resulting in 113 complete moles, 141 partial moles and 189 non-molar abortus. This group concluded that (a) partial mole is about twice as common as complete mole; (b) that molecular techniques are essential for accurate diagnosis in many suspected placental molar cases and (c) they identified aneuploidy in about one-fifth of non-molar cases [8].

Among the 26 diandric triploidies included in our series, only 14 of them (54%) were identified as partial moles by pathology studies. The six cases reported as normal were reviewed by a senior pathologist and, while 3 of these cases were reclassified to partial moles, the remaining 3 were considered inconclusive. It has to be pointed out that not all the pathology studies were performed by a perinatal pathologist, because sometimes they are carried out by non-experienced pathologists. Histopathologic diagnosis of placental mole is imprecise, even when carried out by experts [8]. It has been described that triploidy correlates with the presence of three or four major diagnostic criteria of partial mole: (a) two populations of villi; (b) enlarged, irregular, dysmorphic villi (with trophoblast inclusions); (c) enlarged, cavitated villi (≥3–4 mm); and (d) syncytiotrophoblast hyperplasia/atypia, commonly in addition to histologic evidence of fetal development [12,13]. However, placental hydropic changes present with an inherent difficulty of classification. Hence, Conran et al., assessed the interobserver reproducibility among three pathologists classifying hydropic placentas as hydropic abortus, partial mole or complete mole based on histology resulting in a low diagnostic concordance (kappa values: 0.10–0.37). Among 39 proven diandric triploid losses, 51% were typical partial moles, 8% were probably early partial moles, 10% were probably fibrotic, “ancient” partial moles, 18% were mildly suggestive of partial mole and 13% had no diagnostic feature to suggest partial mole [14]. More recently, Buza et al., in a comprehensive assessment of histology in correlation with DNA genotyping argued that all traditional morphologic parameters attributed to partial moles were nonspecific and were shared in a comparable proportion by various non-molar conditions, including trisomic gestations and hydropic abortions, concluding that genotyping is now to be considered the gold standard in the confirmation and subtyping of sporadic hydatidiform mole [15].

The ability of the ultrasound assessment of the fetus and the placenta to suspect the origin of triploidy is theoretically high because the two phenotypes have no overlapping features: the diandric phenotype presents a relatively well-grown fetus with microcephaly or normal head circumference and a large cystic placenta, while the digynic phenotype manifests with a growth restricted fetus with relative macrocephaly and a non-cystic placenta [16]. However, reported studies describe a limited ultrasound diagnostic capability. In one study, the parental origin could not be established in 20% (12/67) of the cases [17]. In a large series of more than 600 partial moles, only 26% of them could be diagnosed by ultrasound [18]. Finally, in a third series, only a half (102/182) of partial moles were diagnosed by ultrasound [19]. The main limitation appears to be an early gestational age in which the specific features are still not present. In our study, diandric triploidies were diagnosed at a mean gestational age of 8 weeks, after a pregnancy loss in the vast majority of cases, and therefore most of the typical imaging signs were not present at that stage of the pregnancy.

QF-PCR appeared in the field of prenatal diagnosis to overcome the need to culture fetal cells, and to allow a rapid diagnosis of the typical chromosomal anomalies found prenatally. Two large series with more than 1600 fetuses and 22000 samples demonstrated a 92–98% sensitivity and 100% specificity for chromosomes 21, 18, 13, X and Y aneuploidies, and, in particular, 100% for triploidies [20]. The detection of maternal contamination [21] and the parental origin of triploidies requires parental testing, at least a maternal sample. The success rate of QF-PCR in the diagnosis of molar pregnancies has been shown to be 82% in a large lab series, while in our smaller series this rate was 96% (25/26). Alternatively, methylation-specific multiplex ligation-dependent probe amplification (MS-MLPA) can be used as a method for distinguishing between diandric or digynic triploidy without the necessity for parental sample testing [22]. A recent study by Massalska et al., showed that the effectiveness of MS-MLPA to diagnose the parental origin of triploidy was 94%, where the failure rate was 6.0%, probably due to maternal cell contamination [23].

The main limitation of the present study is that it was conducted retrospectively and that paternal samples were not taken together with maternal samples.

To conclude, our clinical experience in miscarriages demonstrates that when products of conception are available for analysis, QF-PCR is more accurate than pathology studies in the diagnosis of a partial mole, and for this reason it is becoming the new gold standard. However, medical management of early pregnancy loss (when a molar pregnancy is not suspected), which is increasingly preferred by women and doctors, precludes the use of products of conception. In those cases, a routine offer of a chorionic villi sampling previous to uterine content evacuation is an option, that our group is exploring. When molar pregnancy is suspected a D&C is usually recommended.

## Figures and Tables

**Figure 1 diagnostics-11-01811-f001:**
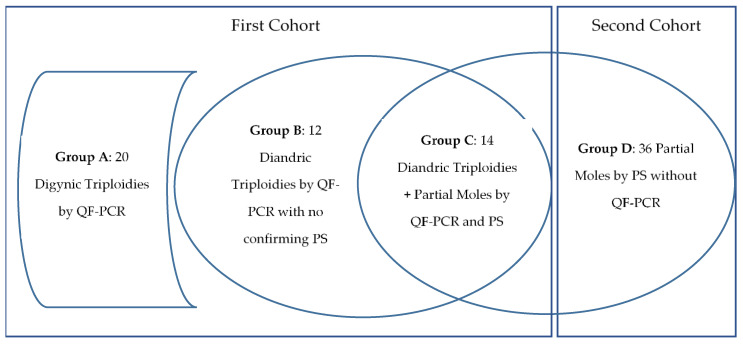
The two studied cohorts: triploidies revealed by QF-PCR after chorionic villi sampling and partial moles assessed by pathologic studies (PS) alone.

**Table 1 diagnostics-11-01811-t001:** Pregnancy characteristics and pathology studies in the 20 digynic and 26 diandric triploidies.

Parameter	Digynic Triploidies	Diandric Triploidies	Comparison between Groups A and B + C
	(Group A)	(Groups B + C)	
	(*n* = 20)	(*n* = 26)	
Mean maternal age (years)	35.2	33.4	*p* = 0.23
Mean β-hCG	69486	97890	*p* = 0.8266
Mean gestational age (weeks)	10.2	8.1	*p* = 0.04
Ongoing pregnancies (*n*; %)	11 (55%)	2 (8%)	*p* = 0.001

**Table 2 diagnostics-11-01811-t002:** Pregnancy characteristics of the two cohorts.

Parameter	Digynic Triploidies	Diandric Triploidies	Partial Moles Only by Pathology Studies	All Triploidies and Partial Moles Studied
	(Group A)	(Groups B + C)	(Group D)	A + B + C + D
	(0)	(*n* = 26)	(*n* = 36)	(*n* = 82)
Mean maternal age (years)	35.2	33.4	33.8	33.9
Mean β-hCG	69,486	97,890	55,649	76,893
Mean gestational age (weeks)	10.2	8.1	8	8.5
Ongoing pregnancies (*n*; %)	11 (55%)	2 (8%)	0	13 (15%)
Partial mole at Pathology studies (*n*)	0	14	36	50 (60.9%)
Pathology studies not available (*n*)	14	6	0	20 (24.4%)
No evidence of partial mole at Pathology studies (*n*)	6	6	0	12 (14.6%)

## Data Availability

The data that support the findings of this study are available from the corresponding.

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
