# Peer review of "The Contribution of QF-PCR and Pathology Studies in the Diagnosis of Diandric Triploidy/Partial Mole"

_diagnostics, 2021, doi:10.3390/diagnostics11101811_

Round 1

Reviewer 1 Report

QF-PCR is NOT a new tool in the diagnosis of triploid pregnancies. A percentage of the pregnancies that are screen positive at the combined first trimester test for aneuploidies is represented by triploid pregnancies; the classical management of such (screen positive) cases is testing by CVS / amnio and use of QF-PCR as the first-line method of genetic analysis. The text should be rephrased to convey the correct message. The paper has merits anyway, does not need to present QF-PCR as a new method of analysis of CVS samples! I would not classify this as major revision, but I think it is an important change to be done by the authors.

The analysis of diandric and digynic triploidies and of miscarriages and ongoing pregnancies together might not be a strength of the study. On the contrary, it does nor help with the already confusingly overlapping classification systems of trophoblastic disease and triploidy. As far as diandric triploidy is concerned, two taxonomies - one genetic (diandric triplody) and one anatomopathological (partial mole) are in use. The concordance of the two taxonomic systems remains to be studied. Although indirect, the contribution of this paper is of interest, in this respect. I think this paper is worth publishing after revision.  

Author Response

QF-PCR is NOT a new tool in the diagnosis of triploid pregnancies. A percentage of the pregnancies that are screen positive at the combined first trimester test for aneuploidies is represented by triploid pregnancies; the classical management of such (screen positive) cases is testing by CVS / amnio and use of QF-PCR as the first-line method of genetic analysis. The text should be rephrased to convey the correct message. The paper has merits anyway, does not need to present QF-PCR as a new method of analysis of CVS samples! I would not classify this as major revision, but I think it is an important change to be done by the authors.

We have added a new sentence in the Introduction to clarify that QF-PCR is not a new tool in the diagnosis of triploidies, although the investigation of the parental origin of the extra set of chromosomes is not a common practice “Although QF-PCR is not a new tool in the diagnosis of triploid pregnancies, as the use of QF-PCR as the first-line method of genetic analysis after chorionic villi sampling or amniocentesis is well established in pregnancies at high risk for chromosomal anomalies, the investigation of the origin of the extra chromosome endowment is not a common practice.”

The analysis of diandric and digynic triploidies and of miscarriages and ongoing pregnancies together might not be a strength of the study. On the contrary, it does nor help with the already confusingly overlapping classification systems of trophoblastic disease and triploidy. As far as diandric triploidy is concerned, two taxonomies - one genetic (diandric triplody) and one anatomopathological (partial mole) are in use. The concordance of the two taxonomic systems remains to be studied. Although indirect, the contribution of this paper is of interest, in this respect. I think this paper is worth publishing after revision.  

We agree with the reviewer. We have added the following sentence to the beginning of the Discussion “As far as diandric triploidy/partial mole is concerned, two taxonomies - one genetic (diandric triploidy) and one anatomopathological (partial mole) are in use. This study was conducted to clarify the concordance between the two taxonomic systems.”

Reviewer 2 Report

It is a manuscript about the usefulness for improving the diagnosis of a partial hydatiform mole by using QF-PCR

The manuscript is very confusing, English must be revised .  

  • The objective (lines 13-14, and 61-62) : to determine the roles of QF-PCR and pathology ? and QF-PCR as a new molecular tool ?
  • The Authors must read carefully the Instructions for the Authors : Improving the diagnosis …..by use is not exactly
  • The histopathologic diagnosis of placental mole, even by experts, is imprecise, I suggest the article by: Colgan TJ et al.   Molecular Diagnosis of Placental Hydatidiform Mole: Innovation and Outcomes, 2017.
  • The references are old and not relevant.
  • It is not very clear what this study found, what is your novelty on a sample non-representative.
  • Introduction: may state the problem and need to offer what is already know – means about the diagnostic method used for the diagnosis of partial hydatiform mole.
  • Line 46 : “identification of molar pregnancies is crucial because they are not viable” if not viable, why is crucial, and why cannot be identified?
  • Line 48 : risk for persistent GTD to be not confounded with the risk of neoplasia , for partial mole is less than 5%.
  • Methods : I suggest headings : Samples , QF-PCR , Karyotype, Histopathology analysis and a table with a complete definition of every step
  • Line 67 : Did the authors perform early amniocentesis ?
  • Lines 79-82 : I suggest to describe the process also the preparation of DNA and of biopsied material and to precise the fluorochromes used .
  • Lines 109-111 : Do you use statistics for which parameters ?
  • Results: a sample of 46 cases is not significant
  • I suggest a table comparing age of gestation / ultrasound signs/ invasive method/ viable-non-viable pregnancy/ outcome /pathology report
  • Lines 155 : 56 partial moles? When ?
  • Discussion may be focused on your objective – do your conclusions answer your objective?
  • You may place your results and compare them with other studies , but regarding QF-PCR and partial mole .

Author Response

It is a manuscript about the usefulness for improving the diagnosis of a partial hydatidiform mole by using QF-PCR

The manuscript is very confusing, English must be revised .

The manuscript has been clarified with the use of the 4 groups defined in Figure 1. The English has been revised

  • The objective (lines 13-14, and 61-62) : to determine the roles of QF-PCR and pathology ? and QF-PCR as a new molecular tool ?

Lines 13-14: The objective has been changed by The aim of our study was to assess the contribution of quantitative fluorescent polymerase chain reaction (QF-PCR) and pathology studies in the diagnosis of diandric triploidies/partial hydatidiform moles.” We have deleted the sentence on QF-PCR as a new molecular tool

  • The Authors must read carefully the Instructions for the Authors : Improving the diagnosis …..by use is not exactly

We have changed the title to “The contribution of QF-PCR and pathology studies in the diagnosis of diandric triploidy/partial mole

  • The histopathologic diagnosis of placental mole, even by experts, is imprecise, I suggest the article by: Colgan TJ et al.   Molecular Diagnosis of Placental Hydatidiform Mole: Innovation and Outcomes, 2017.

Thank you for your suggestion. We have mentioned this study in the Introduction Although the use of QF-PCR has been previously reported by large labs in the diagnosis of molar pregnancies, we are not aware of clinical studies on the same issue” and in the Discussion “A large study conducted in a Canadian lab analyzed over 400 cases of suspected placental moles resulting in 113 complete moles, 141 partial moles and 189 non-molar abortus. This group concludes that partial mole is about twice as common as complete mole and that molecular techniques are essential for accurate diagnosis in many suspected placental molar cases and identified aneuploidy in about one-fifth of non-molar cases

  • The references are old and not relevant.

We have added two recent references, Colgan et al.,2017 and Buza et al., 2021, both commented on in the Discussion.

  • It is not very clear what this study found, what is your novelty on a sample non-representative.

Our clinical experience in miscarriages demonstrates that when products of conception are available for analysis, QF-PCR is more accurate than pathology studies in the diagnosis of a partial mole. However, the medical management of early pregnancy loss (when a molar pregnancy is not suspected), which is increasingly preferred by women and professionals, precludes the use of products of conception. In those cases, a routine offer of a chorionic villi sampling previous to uterus evacuation is an option that our group has explored. When a molar pregnancy is suspected a D&C is usually recommended.

  • Introduction: may state the problem and need to offer what is already know – means about the diagnostic method used for the diagnosis of partial hydatidiform mole.

The last 2 paragraphs of the Introduction have been rewritten

  • Line 46 : “identification of molar pregnancies is crucial because they are not viable” if not viable, why is crucial, and why cannot be identified?

We have replaced the sentence with a new one: “Identification of molar pregnancies is crucial because they involve a higher recurrence risk for future pregnancies and because there is a maternal risk for life-threatening persistent gestational trophoblastic disease, up to 5% after a partial molar pregnancy”

  • Line 48 : risk for persistent GTD to be not confounded with the risk of neoplasia , for partial mole is less than 5%.

The sentence has been rewritten

  • Methods : I suggest headings : Samples , QF-PCR , Karyotype, Histopathology analysis and a table with a complete definition of every step

Headings as suggested

  • Line 67 : Did the authors perform early amniocentesis ?

No, we did not. The earliest amniocentesis was performed at 16 weeks

  • Lines 79-82 : I suggest to describe the process also the preparation of DNA and of biopsied material and to precise the fluorochromes used .

We have added two paragraphs describing this process:

“Genomic DNA was extracted from CV samples using the standard extraction protocol from QIAamp DNA Mini Kit (Qiagen, Hilden, Germany). For amniotic fluid and maternal buccal washes between 0.5 and 2 ml of sample were used, and DNA extraction was performed using the Instagene Matrix (Bio-Rad Laboratories Inc., Hercules CA), following the manufacturer’s instructions.” and “QF-PCR was performed using the Devyser Compact kit (Devyser, Sweden), following manufacturer’s instructions. The kit amplifies 26 highly informative markers from chromosomes 13, 18, 21, X and Y.  PCR products were analyzed with an ABI3130XL (Applied Biosystems, Foster City, CA) and GeneMapper v3.5 software was used to analyze the results. Peak area ratios between 0.8 and 1.4 were considered normal, whereas ratios above and below these were interpreted as trisomy, and the presence of three alleles of equal peak area was also considered trisomy. The presence of a single peak was considered uninformative and a minimum of two concordant informative markers was required to give a result.”

  • Lines 109-111 : Do you use statistics for which parameters ?

We have specified the parameters included in the statistical analysis, including a specific Statistical analysis heading in the Methods, modifying the previous paragraph as followed:

Stata statistical software v.15 was used for statistical analysis of maternal age, gestational age, viability of pregnancy, type of sampling, type of management, parental origin and β-hCG level. Continuous variables (maternal age, gestational age and β-hCG level) were checked with Shapiro-Wilk test and described by mean and SD after verifying they follow a normal distribution. Categorical variables (viability of pregnancy, type of sampling, type of management, parental origin) were described as number of observations and relative frequency. A p-value of <0.05 was considered significant.”

  • Results: a sample of 46 cases is not significant

This study includes two cohorts with 82 partial moles/diandric triploidies overall. The first cohort includes 46 triploid pregnancies diagnosed by QF-PCR with or without pathology studies (groups A+B+C) and the second cohort includes 36 partial moles with pathology studies alone (group D). Now, the two cohorts are better explained in the Methods and Results

  • I suggest a table comparing age of gestation / ultrasound signs/ invasive method/ viable-non-viable pregnancy/ outcome /pathology report

We have added a new table (Table 2) with the pregnancy characteristics of the two cohorts.

  • Lines 155 : 56 partial moles? When ?

Adding all group C (n=14) and group D (n=36) cases, together with the non-confirming pathology studies of group B (n=6), you obtain 56 partial moles. This is now explained in the manuscript.

  • Discussion may be focused on your objective – do your conclusions answer your objective?

Our objective was to assess the contribution of QF-PCR and pathology studies in the diagnosis of diandric triploidy/partial mole. We conclude with the following paragraph: Our clinical experience in miscarriages demonstrates that when products of conception are available for analysis, QF-PCR is more accurate than pathology studies in the diagnosis of a partial mole, and for this reason it is becoming the new gold standard. However, medical management of early pregnancy loss (when a molar pregnancy is not suspected), which is increasingly preferred by women and professionals, precludes the use of products of conception. In those cases, a routine offer of a chorionic villi sampling previous to uterus evacuation is an option, that our group has explored. When molar pregnancy is suspected a D&C is usually recommended.”

  • You may place your results and compare them with other studies , but regarding QF-PCR and partial mole .

In this regard, a new sentence has been included in the discussion: “The success rate of QF-PCR in the diagnosis of molar pregnancies has been shown to be 82% in a large lab series, while in our smaller series this rate was 96% (25/26).

Reviewer 3 Report

Practical aspects of applying QF-PCR to help distinguish 2 types of triploidy and thus to diagnose partial hydatidiform mole were presented in a clinical setting. Also pathological difficulties to detect the condition were highlighted. This is an interesting study with useful information for prenatal specialists.

As authors mentioned another method (MS MLPA) which can distinguish source of extra set of chromosome without parents' sampling they may want to compare it with QF-PCR and discuss wider the differences.

In line 158 indirect writing might be more relevant.

In line 208 there is small grammar error.

There are repetitions of information between Results and Discussion sections which might be left in discussion only.

Author Response

Practical aspects of applying QF-PCR to help distinguish 2 types of triploidy and thus to diagnose partial hydatidiform mole were presented in a clinical setting. Also pathological difficulties to detect the condition were highlighted. This is an interesting study with useful information for prenatal specialists.

As authors mentioned another method (MS MLPA) which can distinguish source of extra set of chromosome without parents' sampling they may want to compare it with QF-PCR and discuss wider the differences.

We have added a new sentence in the Discussion “A recent study by Massalska et al., showed that the effectiveness of MS-MLPA to diagnose the parental origin of triploidy was 94%, where the failure rate was 6.0%, probably due to maternal cell contamination

In line 158 indirect writing might be more relevant.

We have now changed this paragraph in the manuscript.

In line 208 there is small grammar error.

Changed as suggested

There are repetitions of information between Results and Discussion sections which might be left in discussion only.

Some redundant data have been deleted from the Results section

Round 2

Reviewer 2 Report

First of all,  I want to clarify the definitions I use :

  • Gestational trophoblastic disease (GTD) is a group of rare diseases in which abnormal trophoblast cells grow inside the uterus after conception.
  • Hydatidiform mole (HM) is the most common type of GTD= BENIN .
  • Gestational trophoblastic neoplasia (GTN) is a type of gestational trophoblastic disease (GTD) that is almost always malignant.=MALIGN
    • Invasive moles
    • Choriocarcinomas
    • Placental-site trophoblastic tumors
    • Epithelioid trophoblastic tumors

Triploidy is not synonym with GTD .

Persistent trophoblastic disease and choriocarcinoma are very rare pregnancy-related tumours known as gestational trophoblastic tumours (GTTs). In most molar pregnancies, any remaining abnormal tissue in the womb usually dies off , and is spontaneous eliminate.  But , only a very few cases may appear persistent trophoblastic disease. Clinical symptoms – metrorrhagia is the main  sign after a spontaneous abortion leading to the diagnosis also based on the persistence of βHCG level .

Persistent trophoblastic disease is not a life-threatening condition if it resolves spontaneously ! (lines 47-48 , 204-205) and genetic tests are not crucial and genetic tests cannot prevent GTD or GTN ! 

 Lines 47-48 : “Identification of molar pregnancies is crucial because there is a maternal risk for life-threatening persistent gestational trophoblastic disease, up to 5% after a partial molar 48 pregnancy” please provide reference for 5%  

because lines 203-205 you say : Diandric triploidy is associated with partial hydatidiform mole and its diagnosis is crucial to prevent the development of life-threatening persistent trophoblastic disease which accounts for 3.7% of the cases, according to a large review that included 2651 partial

Author Response

We completely agree with you and we are very sorry that the paper was not clear enough to define those concepts.

We changed lines 47-48 to: 

"Identification of molar pregnancies is crucial because they carry a 3.7% maternal risk for persistent gestational trophoblastic disease after a partial molar pregnancy”

And also 203-205:

"Diandric triploidy is associated with partial hydatidiform mole and its diagnosis is crucial to identify persistent trophoblastic disease which accounts for 3.7% of the cases, according to a large review that included 2651 partial..."